# Real-time monitoring and feedback to improve shared decision-making for surgery (the ALPACA Study): protocol for a mixed-methods study to inform co-development of an inclusive intervention

Christin Hoffmann [ID],[1] Kerry N L Avery [ID],[1] Rhiannon C Macefield [ID],[1] Val Snelgrove,[2] Jane M Blazeby [ID],[1] Della Hopkins,[3] Shireen Hickey,[4] Christie Cabral,[5] Jennifer Hall [ID],[4] Ben Gibbison [ID],[6] Leila Rooshenas,[1] Adam Williams,[3] Jonathan Aning,[7,8] Hilary L Bekker,[9,10] Angus G K McNair [ID],[1,3] ALPACA Study team

For numbered affiliations see end of article.

**Correspondence to**
Dr Christin Hoffmann;
c.hoffmann@bristol.ac.uk

## ABSTRACT

**Introduction** High-quality shared decision-making (SDM) is a priority of health services, but only achieved in a minority of surgical consultations. Improving SDM for surgical patients may lead to more effective care and moderate the impact of treatment consequences. There is a need to establish effective ways to achieve sustained and large-scale improvements in SDM for all patients whatever their background. The ALPACA Study aims to develop, pilot and evaluate a decision support intervention that uses real-time feedback of patients' experience of SDM to change patients' and healthcare professionals' decision-making processes before adult elective surgery and to improve patient and health service outcomes.

**Methods and analysis** This protocol outlines a mixed-methods study, involving diverse stakeholders (adult patients, healthcare professionals, members of the community) and three National Health Service (NHS) trusts in England. Detailed methods for the assessment of the feasibility, usability and stakeholder views of implementing a novel system to monitor the SDM process for surgery automatically and in real time are described. The study will measure the SDM process using validated instruments (CollaboRATE, SDM-Q-9, SHARED-Q10) and will conduct semi-structured interviews and focus groups to examine (1) the feasibility of automated data collection, (2) the usability of the novel system and (3) the views of diverse stakeholders to inform the use of the system to improve SDM. Future phases of this work will complete the development and evaluation of the intervention.

**Ethics and dissemination** Ethical approval was granted by the NHS Health Research Authority North West-Liverpool Central Research Ethics Committee (reference: 21/PR/0345). Approval was also granted by North Bristol NHS Trust to undertake quality improvement work (reference: Q80008) overseen by the Consent and SDM Programme Board and reporting to an Executive Assurance Committee.

## STRENGTHS AND LIMITATIONS OF THIS STUDY

⇒ A mixed-methods study design will use a diverse representative sample of surgical patients from a range of National Health Service trusts to determine the feasibility of data collection, the usability of the novel system and understand views of diverse stakeholders to inform use of the system.

⇒ Recruitment will focus on recognised underserved groups (economically disadvantaged, older age, ethnic minority) from Bristol and Bradford to maximise reach to an ethnically and socioeconomically diverse population.

⇒ The study uses three validated questionnaires to monitor shared decision-making (CollaboRATE, SDM-Q-9, SHARED-Q10), including first use of the SHARED-Q10 measure in a surgical setting.

⇒ This study excluded patients without decisional capacity due to distinct requirements and guidance for consent and shared decision-making processes in this population.

**Trial registration number** ISRCTN17951423; Pre-results.

## INTRODUCTION

Shared decision-making (SDM) is a process where patients are supported to reach decisions in collaboration with health professionals.[1] Global and United Kingdom (UK) policy,[2–4] professional and regulatory guidelines[5 6] recommend SDM in all healthcare settings. Making good decisions is particularly important for the 5 million people per year deciding to have surgery in the UK because, unlike many medical therapies, the effects are usually immediate and irreversible. Ensuring patients and surgeons

have discussed accurate information about all options and their consequences, exchanged their reasoning about, and preferences for, each option, and agreed the treatment plan is essential to a good SDM process.

Evidence shows there is scope to improve SDM for surgery. A systematic review of 22 surgical studies found that only 36% of 13 176 patients perceived their consultation as shared.[7] Other systematic reviews show that surgeons underestimate patients' information needs,[8] and patients do not receive desired information before surgery.[9] Major surgical risks go undisclosed,[10] and patients report feeling uninformed[11] and want more involvement in decision-making.[12] The impact of these deficiencies is inadequately understood. It is thought that improving SDM processes may lead to more effective care through enhanced clinician–patient reasoning,[13] thereby supporting treatment choices with greater benefit/harm ratios[8] and reducing overall use of health services.[14 15] High-quality SDM may also moderate the impact of treatment harms through more realistic treatment expectations[16 17] and improved self-management.[18]

Guidelines for the implementation of SDM have been recently published by the National Institute for Health and Care Excellence (NICE) that includes best evidence from a Cochrane review[9] and consultation with 454 stakeholders. It concluded that a combination of interventions to support organisations, clinicians and patients is needed, but the evidence for these interventions is often poor.[19] Key priority areas were identified for future research, including generating evidence about how to: (1) sustain SDM implementation at an organisation/health service level, (2) measure the effectiveness of the SDM process for different contexts/settings/people, and (3) ensure the SDM process is inclusive of people from diverse backgrounds (eg, ethnic minorities, persons of lower health literacy or income backgrounds).

The ALPACA Study aims to address these deficiencies. We will develop, pilot and evaluate a decision support intervention that uses real-time feedback of patient experiences of the SDM process to impact patient and professional decision-making processes before adult elective surgery and improve patient and health service outcomes. The intervention will include (1) efficient, real-time evaluation of patient experiences of SDM at scale, (2) timely feedback of individual patient-reported experiences of SDM to care teams before surgery and (3) activities supporting meaningful change in patient and professional decision-making about surgery, individually and together.

This project aims to enable surgical teams to remedy deficiencies in the SDM process before surgery and thereby addresses NICE research priorities to detect such deficits reported by the patient. The intervention will be deliverable at scale to create sustained improvement in SDM through system-wide changes in decision-making processes facilitated by continuous patient-reported feedback. It will be co-created with patients with a focus on inclusivity of recognised underserved groups. Developing methods for efficient evaluation of the SDM process will make measurement of SDM outcomes more consistent and meaningful.

## Aim and objectives
The overall aim of this project is to develop, pilot and evaluate a decision support intervention that uses real-time feedback of patient experience of SDM to change patient and professional decision-making processes before adult elective surgery and improve patient and health service outcomes. There are three phases with the following objectives:

Phase 1: assess the feasibility, usability and stakeholder views of implementing an automated system to monitor the SDM process for surgery in real time.

Phase 2: co-develop and refine the intervention with patients and professionals to understand how the intervention works, for whom and in what context using findings from phase 1.

Phase 3: evaluate the effectiveness, cost-effectiveness and implementation of the intervention to improve patient and health service outcomes in the English National Health Service (NHS).

This protocol describes phase 1. Details of subsequent phases which will complete the development (phase 2) and evaluation of the intervention (phase 3) will be described in future publications.

## METHODS
The project will employ mixed-methods to develop a complex intervention comprising multiple components that will impact a wide range of stakeholders and system processes. The overall aim to develop and evaluate the intervention will be conducted according to Medical Research Council (MRC) guidelines.[20] Phase 1 reported here is consistent with the MRC framework's feasibility phase, with consideration of the core elements critical for complex intervention research. Any qualitative elements will be reported in accordance with the Consolidated criteria for Reporting Qualitative research guidelines.[21]

## Conceptualisation
There is no unified definition of SDM. A systematic review identified 40 SDM models currently available with 53 different elements clustered in 24 overarching components.[1] Components present in more than half of models were: 'describe treatment options' (88% of models); 'make a decision' (75%); 'patient preferences' (68%); 'tailor information' (65%); 'deliberate' (58%); 'create choice awareness' (55%) and 'learn about the patient' (55%).

This study will conceptualise SDM using the 'Three Talk model' (2012),[22] later refined to 'Implement-SDM' (2019).[23] This single-component model provides a

guide for enhancing health professional communication to deliver SDM, and is the most highly referenced model (>1800 citations, Web of Science). It involves three key steps consistent with other models of SDM: (1) introduction of choice, (2) describing options and (3) helping patients explore preferences and make decisions. This was the chosen model for an NHS MAGIC Programme[7] and is recommended in NICE guidelines.[24]

## Setting

Research will be conducted at three UK hospital trusts (North Bristol Trust/NBT, University Hospitals Bristol and Weston NHS Foundation Trust/UHBWFT and Bradford Teaching Hospitals NHS Foundation Trust/ BTHFT) alongside quality improvement programmes to improve SDM. NBT is one of the largest acute NHS trusts in the UK.[25] It provides a full range of acute clinical care for both local and regional clinical commissioning groups in South-West England. Specialised services are provided through NHS England, Welsh Health Boards and Welsh Specialist Commissioners. Services provided include elective and emergency gastrointestinal surgery, obstetrics and gynaecology, as well as specialist regional services in urology, neurosciences, trauma and orthopaedic and vascular surgery. One UHBWFT department is included as the South-West England regional cardiac surgical centre. BTHFT is an acute trust in the North of England with a full range of elective and emergency surgical services. Bristol and Bradford were purposively selected to maximise reach to ensure a diverse representative sample is included (eg, 26.8% classed as Asian or Asian British, compared with 5.5% in Bristol).

## Phase 1: assess the feasibility, usability and stakeholder views of implementing an automated system to monitor the SDM process for surgery in real time

Phase 1 will determine whether it is feasible and acceptable to monitor SDM processes for surgery automatically and in real time using a novel electronic system. Objectives are to explore:

1.1 Feasibility of automated data collection.
1.2 Usability of the electronic measurement system.
1.3 Views of diverse stakeholders to inform the use of the system to improve SDM.

Each objective comprises separate methods which are described in turn below. This phase is expected to continue until June 2025.

## Feasibility of automated data collection

Feasibility assessment is designed to establish the feasibility of automated real-time evaluation of patient experiences of SDM at scale and will identify opportunities to optimise recruitment and data collection.

## Participants

All patients over the age of 18 years who have been booked for planned vascular, gastrointestinal, urological, neurosurgical, gynaecological, breast, cardiac and orthopaedic surgical procedures at participating hospitals will be eligible to participate. Surgical departments have been selected to be broadly representative of a diverse range of surgical specialties. Excluded will be patients under the age of 18 years, those without decisional capacity to consent for medical procedures, or undergoing unplanned (emergency) surgery or endoscopic procedures. Data related to eligibility criteria are routinely collected through electronic patient record (EPR) systems.

## Measurement of patient experience

Real-time measurement of patient experiences of the process of SDM will be facilitated by a secure, automated system procured through a third-party provider approved by NHS trusts. The system is a customisable off-the-shelf electronic patient-reported outcome measurement software and has previously been used for electronic data capture in other countries. Eligible patients will be identified through EPR using algorithms developed in collaboration with the software provider. Structured data queries will be designed to extract details of patients booked for eligible procedures. Queries will be designed to run automatically, securely transferring data from the hospital to the software provider daily to account for changes in scheduling. The automated system will send three validated SDM measurement instruments within 1 day after surgery booking (real-time baseline measurement). This time point in the decision-making process was chosen as a pragmatic point in time to represent patients' cumulative experiences of SDM for surgery which may include discussions with surgeons, physicians, general practitioners, nurses, family and friends. The selected measurement instruments will be operationalised into an online survey and administered via short messaging service (SMS) or email. A reciprocal data feed will securely return patients' survey responses immediately to the hospital data warehouse for secure storage (real-time analysis and feedback). Follow-up measures are sent within 1 day before surgery by either SMS or email (real-time follow-up measurement). A schematic of the process and intervention aims is illustrated in figure 1.

Selection of three SDM measures was made through discussions within the study team and was informed by a systematic review of SDM measurement instruments using COSMIN (consensus-based standards for the selection of health measurement instruments) methods,[26] national guidelines[19] and recommendations and use within NHS clinical practice.[27–29] The CollaboRATE instrument is a validated three-item patient-reported measure assessing the extent of SDM experienced by patients.[30] Assessment of the instrument using COSMIN methods demonstrated acceptable discriminative validity, concurrent validity, intrarater reliability and sensitivity to change.[26] It has been used in excess of 40 studies,[31] including evaluations of quality improvement projects in surgery.[32] The

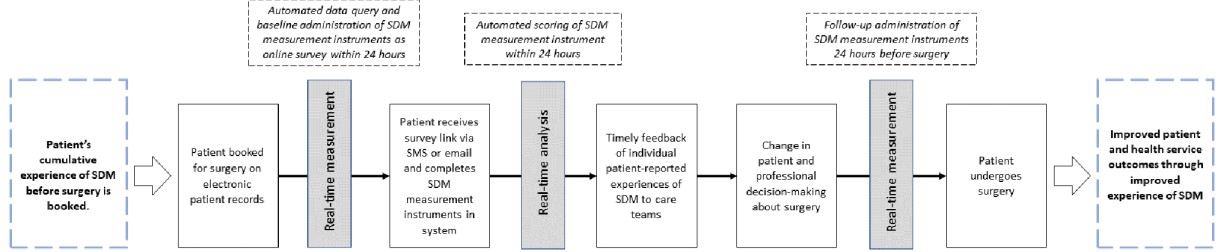

**Figure 1** Schematic of measurement process. SDM, shared decision-making; SMS, short messaging service.

SDM-Q-9 instrument is a validated nine-item patient-reported measure that evaluates their perceptions of involvement in the decision-making process.[33] It has been widely used in interventional studies and demonstrates good reliability and structural validity.[26 33] Systematic review evidence recommended use of SDM-Q-9 for surgery.[34] The SHARED-Q10 instrument is a 10-item patient-reported measure to assess patient perceptions of information provided, involvement in consultations and agreement with the decision made.[35] This measure is included because it was developed, validated and used in an NHS quality improvement programme[36 37] and evaluates domains beyond patient perception of professional communication. Complete measurement instruments can be found in online supplemental figures 1–3.

## Analysis

Feasibility of real-time monitoring will be evaluated by analysis of overall recruitment rate, response rates and time to response for the SDM measures at baseline and follow-up. Response rates will be presented as a number and percentage based on patients who completed the measures (eg, completed all three items of the CollaboRATE instrument). Issues of equality, diversity and inclusion will be explored by examining the correlation between responders/non-responders and sociodemographic patient variables extracted from EPR.

Relationships between responders and non-responders and clinical and sociodemographic details will be explored using univariable and multivariable logistic regression. Included will be age, sex, ethnicity, Index of Multiple Deprivation, and clinical and treatment parameters (eg, operation (three-digit Classification of Interventions and Procedures (OPCS) code), diagnosis (International Statistical Classification of Diseases and Related Health Problems (ICD-10) code), date of booking, specialty, number of outpatient appointments in relevant specialty, number and length of hospital inpatient episode in relevant specialty). Index of Multiple Deprivation will be derived by lower layer super output area for individuals' postcode. All variables will be extracted from routinely collected data in EPR.

In addition, the study team will document any relevant technical, financial, administrative and logistical observations throughout the study and pertinent

challenges using shared electronic records (eg, Microsoft Office suite). Any learning points will be descriptively summarised.

## Usability of the electronic measurement system

Post-deployment usability testing will be conducted according to International Standards Organisation (ISO) standards for human–systems interaction (9241-11:2018) to evaluate the system's use in this context.[38 39] System users, defined as anyone who is a current or prospective surgical patient, will be invited to participate in a mixed-methods usability evaluation to assess system effectiveness, system efficiency and user satisfaction (see box 1). To complete usability evaluation, a process map will be created to define the number and type of task required to complete the measurement system.

## System effectiveness

One-to-one user testing sessions will be used to assess system effectiveness by evaluating task completion and error rates. Sessions will involve completing the automated system in a simulated environment, applying concurrent think aloud techniques.[40–42] A topic guide will be developed and will structure the testing session discussions (online supplemental table 1).

Patient and public representatives will be invited to participate in online user testing sessions. They will be eligible if they are over the age of 18 years. Individuals from two patient experience panels (NBT, BTHFT) will be recruited through respective panel coordinators. Sampling will be purposive to maximise variation in geographical location, ethnicity and sex, and will aim to include individuals whose first language is not English.

---

**Box 1    Definition of usability concepts**

⇒ *System effectiveness:* the ability of participants to perform tasks to achieve predetermined goals completely and accurately, and without negative consequences (eg, poor layout of the system interface leading to participants missing or accidentally selecting system options).[38 39 58 59]

⇒ *System efficiency:* the amount of participant resources required to achieve the prespecified goals (eg, system completion time).[59 60]

⇒ *User satisfaction:* the subjective opinions of participants based on their experience interacting with the system.[59] This includes any subjective reports about likes, dislikes and recommendations for changes.[38]

---

User testing will be completed using video-conferencing software (eg, Zoom, Microsoft Teams) and audio-recorded. Two researchers familiar with the automated system and trained in qualitative research will conduct the user testing sessions. Observational notes will be taken to collect further information about challenges or errors encountered during task completion.[43 44]

Task completion rates will be calculated as percentage of tasks completed. Error rates will be calculated based on number of user errors encountered. User errors will be deviations or problems experienced that will interfere with successfully completing the task. Number and type of non-critical errors (successfully addressed by testers themselves following instructions from the observer) and critical errors (require the observer to intervene or take remedial actions) will be noted. Results will be presented using descriptive statistics.

Understanding of system effectiveness will be supplemented by analysis of response rates generated through feasibility work in 1.1.

### System efficiency

System efficiency will be assessed by calculating task completion time and task efficiency. Task completion time is defined as the time participants took from the first activity (starting the survey by following the hyperlink) to the last activity (submission of the survey). Task efficiency is defined as the time spent to complete each task. Analyses will be based on those who completed the automated system and for whom first and last activity timestamps were available.

### User satisfaction

One-to-one user interviews will be conducted to assess user satisfaction in depth. Interviews will explore issues including ease of use/navigation, satisfaction with instructions, satisfaction with the visual display, ease of access, burden and likelihood of using the system again. Barriers and facilitators to completing the measurement system will also be explored. A topic guide will be tested and refined and used to direct discussions.

A subset of eligible patients and participants of the user testing sessions will be invited to take part. A purposive sampling strategy will be adopted to ensure that insights are drawn from a range of perspectives. Sampling characteristics will be (1) experience with surgery (vascular, gastrointestinal, urological, neurosurgical, gynaecological, breast or orthopaedic surgery) and good/bad SDM experience, (2) sex, (3) age, (4) ethnicity and (5) individuals whose first language is not English. Participant characteristics will be assessed as the study progresses and recruitment efforts will focus to target under-represented patients as necessary. Recruitment of the subset of patient participants will be undertaken by the principal investigator, research nurse or clinical collaborators via email or telephone. User testing participants will be recruited by researchers during the user testing sessions and interviews

will be conducted immediately following the user testing session.

Interviews will be conducted primarily remotely (eg, telephone or video conference) by experienced and trained qualitative researchers. All audio-recorded interviews will be transcribed and anonymised. Transcripts will be thematically analysed (see below).

### Views of diverse stakeholders to inform the use of the system to improve SDM

Qualitative research with a wide range of stakeholders (including patients, healthcare professionals and members of the community) will be conducted to understand views of multiple stakeholders to inform the use of the system to improve SDM. Opinions about the acceptability and potential impact of real-time monitoring of SDM will be sought. Views on potential intervention components (activities), mechanisms of change, intermediate outcomes, assumptions and indicators will be explored. Results will be used to co-develop initial programme theory to inform phase 2.

Patients and members of the public and community over the age of 18 years will be eligible to take part. The sample will include people who are disproportionately affected by a poor SDM process and outcomes of surgery: those who are economically disadvantaged, from minority ethnic groups and in older age.[45–48] Professionals working in participating trusts will be eligible for inclusion and may include surgeons, anaesthetists, nurses, perioperative care physicians, allied health professionals and hospital managers.

### Recruitment

Eligible participants will be identified through existing networks, collaborations with local hospital patient panels, community leaders and patients who have participated in feasibility (1.1) and usability (1.2) data collection. We will seek to recruit individuals who experience multiple intersecting inequalities to ensure the views of those with barriers to accessing healthcare are incorporated.[45] Recruitment of members of the community will be conducted using techniques developed and successfully applied by the Born in Bradford team[49 50] and the patient and public involvement and engagement (PPIE) group of the National Institute for Health and Care Research Bristol Biomedical Research Centre. Recruitment materials will be translated into most spoken languages within the local areas.

Purposive sampling will seek to achieve diversity in relation to sociodemographic characteristics (eg, age, gender), experience with surgery or SDM (eg, surgical specialty, good/bad SDM experience) or underserved groups (economically disadvantaged, older age, ethnic minority). Where appropriate, snowball sampling will also be used, whereby individuals who participate in the study are asked about other potentially interested participants. The sample size will ultimately depend on theoretical saturation (eg, when no new insights are

identified from the data and sufficient data are collected to address the research question).[51 52] It is anticipated that approximately 130 participants (around 105 patients and members of the community, and 25 professionals) will be required.

## Data collection

Data collection will apply a flexible strategy to minimise perception that the research is 'hard to engage with'.[49] A range of qualitative research methods are planned remotely and/or face-to-face including semi-structured interviews, focus groups and participatory approaches (eg, community events, discussion groups). It is anticipated that a minimum of 30 one-to-one interviews and 6 focus groups are required, complemented by recruitment through community events and discussion groups. However, these methods may be adapted based on evolving best-practice evidence from citizen science[53] and feedback from PPIE stakeholders. For example, evidence suggests that some British Asian people may be more willing to participate in a focus group in a familiar setting (eg, community centre) than other settings.[54]

Interviews and focus groups will be facilitated by experienced qualitative researchers based in Bristol and Bradford. Topic guides for interviews and focus groups will be developed to direct discussions. This will be iteratively refined during data collection to explore emergent views. Interviews and focus groups will be held face-to-face, over the telephone or using a secure video-conference service (eg, Zoom, Microsoft Teams) but will ultimately depend on participant preference. Data collection will primarily be conducted in English. However, where data are collected from non-English speaking members of the community, additional support will be provided by interpreters and specialist researchers who conduct relevant foreign language interviews and focus groups. All interviews and focus groups will be audio-recorded and transcribed verbatim. Field notes will be taken during the interviews.

## Qualitative analyses

Transcripts and field notes will be analysed using a thematic approach with the help of qualitative data management software (NVivo). Principles of thematic analysis will be applied to the data whereby (1) transcripts and notes will be read and reread, (2) codes are generated and assigned to relevant excerpts within the transcripts, (3) themes will be identified by collating similar codes, (4) accuracy of themes will be checked and (5) detailed analysis of themes will take place.[55] Analysis will involve linking transcripts and observational notes by integrating relevant data from both sources to gain a more comprehensive understanding of key findings. This process will primarily be inductive, with codes developed and iteratively refined through interpretation of the data. There will, however, be an a priori interest in examining data in relation to the study aims. For example, information to

support evidence for the acceptability of monitoring the SDM process and impact of monitoring on clinical care will actively be sought.

Analyses will be conducted separately for different stakeholder groups (patients, professionals, community) to help ascertain different viewpoints or experiences reported by each participant group. Depending on findings, an additional layer of analysis may be conducted to contrast results for several subgroups (eg, different underserved groups; different specialties) to ensure differing perspectives and experiences by population and context are accounted for in later intervention development. At least two experienced qualitative researchers will perform analysis independently and meet regularly to discuss impressions of the data. A subset of transcripts will be double-coded by another experienced qualitative researcher. Any discrepancies in coding or interpretation of data will be referred to the wider study team for further discussions.

Summaries of findings from the analyses (descriptive reports) will be written, combining preliminary findings from the various data sources in relation to the study objectives. Drafts of these summaries will be prepared following rounds of recruitment and analyses and discussed within the study team. The summaries will be iteratively developed as analysis proceeds and will inform discussions about saturation.

Dedicated multidisciplinary meetings involving public contributors will be held to articulate an initial programme theory to inform the future development of the intervention to be more inclusive of recognised underserved groups. A summary of key findings from qualitative data collection will be prepared. We will draw on behavioural (Capability-Opportunity-Motivation Behaviour (COM-B) model)[56] and organisational (Normalisation Process Theory)[57] change theory to identify theory of how the intervention will work for underserved groups. Summaries will be combined to form a comprehensive report, providing a basis for phase 2.

## Data management

All data will be generated and handled in accordance with relevant directives and regulations (eg, Data Protection Act 2018). Any data collected as part of qualitative data collection will be recorded using encrypted devices. Audio files will be securely transferred and transcribed by transcription services approved by the University of Bristol. Transcripts will ensure anonymity of participants (eg, in future study outputs) by assigning pseudonyms or participant IDs to replace any names or identifiable information. All electronic data files will be saved in restricted folders only accessible to the research team, on secure University of Bristol network space that adheres to the University of Bristol's data security policies. Files containing any personal information (eg, contact details) will exclusively use the linked participant

ID and will be encrypted and stored securely on the university servers.

## STUDY STEERING GROUP

A dedicated study steering group will be convened to provide oversight and strategic direction for the study. It will include patients and independent clinical and methodological experts and will meet 6 monthly to review progress and provide strategic guidance.

## PATIENT AND PUBLIC INVOLVEMENT AND ENGAGEMENT

PPIE is central to the project and will play a key role throughout. Patient partners have helped define the research questions and draft the protocol. A PPIE strategy has been developed in collaboration with patient partners in the planning stages of this study to ensure it meets the needs of patients. It includes PPIE activities across (1) strategy and oversight, (2) study conduct and (3) dissemination. Involvement of the patient coauthor (VS), a patient advisory group consisting of members from a diverse background and patient representatives on our steering group will ensure the study focuses on patient needs throughout. PPIE activities will be coordinated by an experienced researcher and will be evaluated. Any feedback will be used to iteratively evolve the PPIE strategy to meet the needs of advancing PPIE practices.

## ETHICS AND DISSEMINATION

This study is part of a project spanning quality improvement and research. It is therefore subject to two governance processes requiring separate approvals: approval to monitor patients' experience of SDM in routine clinical practice was initially approved through a quality improvement proposal at North Bristol NHS Trust (reference: Q80008). This was then incorporated into a larger programme of work, where all processes were approved through the appropriate governance framework (Consent and SDM Programme Board, reporting to an Executive Assurance Committee). Patients will provide consent to participate in real-time monitoring through indicating their agreement with terms and conditions for the programme of work before completing the survey administered through the measurement system.

Ethical approval required to conduct qualitative data collection with NHS patients and professionals was granted by the NHS Health Research Authority North West-Liverpool Central Research Ethics Committee (reference: 21/PR/0345). Participants will provide written consent to participate in qualitative data collection before any research activity will commence. Consent will be obtained electronically through a link to a secure data management platform (REDCap V.11.1.18). As part of the consent process, participants will agree to their anonymised quotes being published in scientific journals.

The results of this work will be presented to professionals (at conferences, as journal articles), shared with the public (social media, engagement events) and those who participated in the project. We will collaborate with organisations involved in SDM (NICE, NHS England) to share findings from the study and maximise the value of our work. Materials produced for dissemination will be tailored to the target audience and will include plain summaries in various languages, formal and informal presentations, infographics or posters.

**Author affiliations**
[1] National Institute for Health and Care Research Bristol Biomedical Research Centre, Bristol Centre for Surgical Research, Bristol Medical School: Population Health Sciences, University of Bristol, Bristol, UK
[2] Patient Representative, Bristol, UK
[3] North Bristol NHS Trust, Bristol, UK
[4] Bradford Institute for Health Research, Bradford Teaching Hospitals NHS Foundation Trust, Bradford, UK
[5] Centre for Academic Primary Care, Bristol Medical School, University of Bristol, Bristol, UK
[6] University Hospitals Bristol and Weston NHS Foundation Trust, Bristol, UK
[7] Bristol Urological Institute, Southmead Hospital, North Bristol NHS Trust, Bristol, UK
[8] Population Health Sciences, Bristol Medical School, University of Bristol, Bristol, UK
[9] Leeds Unit of Complex Intervention Development (LUCID), Leeds Institute of Health Sciences, School of Medicine, University of Leeds, Leeds, UK
[10] The Research Centre for Patient Involvement (ResCenPI), Department of Public Health, Aarhus Universitet, Central Denmark Region, Denmark

**Collaborators** The ALPACA Study team: Andy Judge (National Institute for Health and Care Research Bristol Biomedical Research Centre, Bristol Centre for Surgical Research, Bristol Medical School: Population Health Sciences, University of Bristol, Bristol, UK); Andrew Smith (North Bristol NHS Trust, Bristol, UK); Archana Lingampalli (North Bristol NHS Trust, Bristol, UK); Barnaby Reeves (Bristol Trials Centre, Population Health Sciences, Bristol Medical School, University of Bristol, Bristol, UK); Jessica Preshaw (North Bristol NHS Trust, Bristol, UK); Michael R Whitehouse (National Institute for Health and Care Research Bristol Biomedical Research Centre, Bristol Centre for Surgical Research, Bristol Medical School: Population Health Sciences, University of Bristol, Bristol, UK, North Bristol NHS Trust, Bristol, UK); Paul Cresswell (North Bristol NHS Trust, Bristol, UK); Philip Braude (North Bristol NHS Trust, Bristol, UK); Shelley Potter (North Bristol NHS Trust, Bristol, UK); Timothy Beckitt (North Bristol NHS Trust, Bristol, UK); Timothy Whittlestone (North Bristol NHS Trust, Bristol, UK).

**Contributors** AGKM developed the original idea for this study along with KNLA and JMB. AGKM, KNLA, CH, RCM, JMB, DH, SH, CC, JH, BG, LR, AW, JA and HLB contributed to the development of the research question and objectives and were involved in the design of the study protocol. KNLA, RCM, CC, LR, JH and HLB provide methodological expertise. CH and AGKM wrote the first draft of the manuscript, and all coauthors reviewed and critically appraised the manuscript. AGKM (guarantor) has overall responsibility for the content and project with strategic oversight from JMB. VS contributed to the PPIE strategy. Collaborators part of the ALPACA Study team (AJ, AS, AL, BR, JP, MRW, PC, PB, SP, TB, TW) provide clinical liaison and subject expertise that have shaped the study design. All collaborators have critically reviewed the study proposal. All authors read and approved the final version.

**Funding** This project is funded by the National Institute for Health and Care Research (NIHR) Programme Development Grant (NIHR205174). It is also supported by an NIHR Clinician Scientist award to AGKM (NIHR CS-2017-17-010) and by the NIHR Biomedical Research Centre (BRC) at the University Hospitals Bristol and Weston NHS Foundation Trust and the University of Bristol (BRC-1215-20011, NIHR203315).

**Disclaimer** The views expressed in this publication are those of the authors and not necessarily reflect those of the NIHR, the Department of Health and Social Care or the UK National Health Service.

**Competing interests** None declared.

**Patient and public involvement** Patients and/or the public were involved in the design, or conduct, or reporting, or dissemination plans of this research. Refer to the Patient and public involvement and engagement section for further details.

**Patient consent for publication** Not required.

**Provenance and peer review** Not commissioned; externally peer reviewed.

**ORCID iDs**
Christin Hoffmann http://orcid.org/0000-0002-6293-3813
Kerry N L Avery http://orcid.org/0000-0001-5477-2418
Rhiannon C Macefield http://orcid.org/0000-0002-6606-5427
Jane M Blazeby http://orcid.org/0000-0002-3354-3330
Jennifer Hall http://orcid.org/0000-0001-8379-5555
Ben Gibbison http://orcid.org/0000-0003-3635-6212
Angus G K McNair http://orcid.org/0000-0002-2601-9258

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
