## [Reviewer comments · BMJ Open]

ARTICLE DETAILS

TITLE (PROVISIONAL)	Protocol for a mixed-methods study to inform co-development of an inclusive intervention that uses real-time monitoring and feedback to improve shared decision making for surgery (the ALPACA study)
AUTHORS	Hoffmann, Christin; Avery, Kerry; Macefield, Rhiannon; Snelgrove, Val; Blazeby, Jane; Hopkins, Della; Hickey, Shireen; Cabral, Christie; Hall, Jennifer; Gibbison, Ben; Rooshenas, Leila; Williams, Adam; Aning, Jon; Bekker, Hilary; McNair, Angus; Judge, Andrew; Smith, Andrew; Lingampalli, Archana; Reeves, Barnaby; Preshaw, Jessica; Whitehouse, Michael; Cresswell, Paul; Braude, Philip; Potter, Shelley; Beckitt, Timothy; Whittlestone, Timothy

VERSION 1 – REVIEW

REVIEWER	Bader, Angela Brigham and Women's Hospital
REVIEW RETURNED	05-Sep-2023

GENERAL COMMENTS	I appreciate the opportunity to review this interesting manuscript, however would need more information to accurately comment on the proposed protocol. 1. First, no degrees are listed with the authors. What exactly are the degrees, are any of the authors in clinical departments, and are any of the coauthors anesthesiologists or surgeons? I think clinical expertise [particularly in surgical areas is required among the authors, not just among those interviewed. 2. There needs to be tables and figures of exactly what the survey questions are that are going to be sent, are three different surveys actually going to be sent to patients? will they be combined? I don't think I can comment without an understanding of exactly what is going to be sent to the patients and how the answers are going to be recorded. There is a sentence that states "SDM patient surveys will be set via SMS or email within one day after booking" as well as the day before surgery. In my own institution this would result in an extremely low response rate, have the authors considered this? Have they done any small pilots to see if this would even be feasible? I cannot tell from the methods section exactly what is going to be pulled from the EMR to answer the questions asked. Have the SDM instruments actually been used for surgical patients successfully to obtain the exact information that these authors are seeking? The references quoted dont give this specific information.
--

REVIEWER	Clapp, Justin University of Pennsylvania
REVIEW RETURNED	18-Sep-2023

GENERAL COMMENTS

Thanks for the opportunity to review this manuscript. The authors describe a protocol for measurement of shared decision making in a wide array of patients across 3 NHS trusts as the initial phase of a multi-phase project for increasing the frequency and quality of SDM. This initial phase also involves extensive mixed-methods assessment of the feasibility and acceptability of the authors' strategy for measuring SDM. The protocol is well written and well cited, though I do have a few comments/concerns.

(1) I find myself puzzled by the order of the phases. In 1.1, the authors appear to be launching their SDM measurement approach (consisting of 3 surveys disseminated via phone/email to patients identified through the HER) in order to assess recruitment and response rates. But then in 1.2 the authors describe carrying out usability testing. Wouldn't it be preferable to first carry out usability testing before even figuring out which surveys should be used and how they should be disseminated and formatted? Then in 1.3, the authors describe an extremely extensive plan to obtain feedback from patients and members of the public in part about the "acceptability and potential impact of real-time monitoring of SDM." But again, shouldn't this feedback be obtained prior to extensively launching a set of surveys that have already been decided upon by the investigators?

(2) The sheer amount of work proposed by the investigators to evaluate their measurement approach and possibilities for an intervention must be emphasized. Just the qualitative work in 1.3 would exceed the amount of data in many PhD dissertations in sociology or anthropology. I understand the urge to thoroughly assess feasibility/acceptability/usability and to get the input of diverse stakeholders, but I have doubts that all of this work can be accomplished in a reasonably timely fashion, and I'm really not sure it's all necessary to design a method for pushing out some brief surveys.

(3) This is a big-picture comment, but I wonder about the worthwhileness of all this work given that it's all predicated on the SDM construct. We're a good 30 years into SDM work, and so much remains flimsy: as the authors note, we still lack a clear definition of what SDM is; there isn't much evidence that it can be reliably identified in practice, given that the numerous surveys and observation scales that have been designed to measure it have consistently been shown not to correlate; and there is, likely partially as a result of these problems, little evidence that SDM produces substantial effects on outcomes. I just wonder whether some portion of all this effort should instead be used to try to refine our models of high-quality clinical communication.

(4) Two minor notes:

a. The authors repeatedly state that they will be performing "real-time" measurement of SDM, but I was never able to figure out what exactly that means (i.e., what sets their approach off from any other effort to measure SDM in the clinic).

b. I don't have a clear sense of how the authors will be obtaining consent from the patients whose info is apparently being derived from the EHR so that the surveys can be pushed out to them.

REVIEWER	Sanchez, Sabrina Boston Medical Center
REVIEW RETURNED	19-Sep-2023

GENERAL COMMENTS	Thank you for the opportunity to review this manuscript. First, I would like to commend the authors on a herculean undertaking- the improvement of shared decision making use in surgery is as important and timely as it is challenging. I specially appreciate that the authors are starting small and right at the begging by creating a protocol to evaluate the feasibility, usability, and acceptability of the first steps of an intervention they envision will help increase the use and improve the process of SDM in surgery. They approached their intervention design thoughtfully, methodically, comprehensively, and most importantly, humbly with regards to all the stakeholders involved in the process. I only have some clarifying questions with regards to the protocol that once addressed I think will strengthen the manuscript for thorough understanding by the readers: 1- Page 8, line 51: Patients without capacity to consent for medical procedures will be excluded, how will this be determined? and/or, at what point in the protocol will this be determined? Will these patients receive the SDM instruments to fill out and once it is determined they are unable to consent, be excluded? Or will they be excluded beforehand, through the algorithm developed with the software provider? 2- Page 9, line 15: I am curious about the booking process after a patient sees a surgeon. In order to have this intervention really be a "real time" intervention, it would be important to ensure the SDM questionnaires are received/filled out by the patient shortly after the decision-making conversation with the surgeon (hours to days). It seems like the algorithm will be such that a patient will receive the SDM questionnaires after the surgery is booked in the EPR, so, at the participating hospitals, are surgeries always booked within hours to days of the decision for surgery being made? At many of the hospitals that I know of, that is not the case, and very often cases don't get booked for weeks to months after the decision for surgery is made. In hospitals of this kind the intervention presented would not be "real time". Can you please clarify this point? 3- Page 9, lines 49-51: Will you evaluate the time frame of responses after patients receive the questionnaire? It seems like this may be an important thing to measure with regards to evaluating the feasibility of this project. It may be very different if someone responds within 3 hours of receiving the surveys vs 3 weeks. If you are able to get a 90% response rate but you only get this several weeks after the initial conversation, this may not be as useful as getting a 90% response rate at 3 days, and a 3 week delay would not constitute "real time", while 3 days could 4- On a similar note, do you have an idea yet of what "real time" may mean in the context of this intervention? Will it be a few days, a week, a month? Or is this something that you will gather based on responses once the intervention starts? 5- Page 12, line 54: A reference or two supporting that economically disadvantaged, minority race/ethnic groups, and elderly patients are disproportionately affected by poor SDM processes would be helpful. 6- I eventually understood that you will be rolling out the SDM questionnaires as part of QI and as such, all patients that meet inclusion criteria will be included, there will be no patient consent
--

	required, and there will be no option to not participate as you evaluate the feasibility of automated data collection. While this is explained at the end of the manuscript I spent several pages prior, as I was reading, wondering about consent and opt out ability for patients from even receiving the questionnaires. I feel like it may be helpful to explain this explicitly and early in the manuscript (potentially the "Setting" section). Again, this is an incredibly timely and important step in improving SDM in surgery and I am eager to see the results of this intervention at all stages, from the feasibility, usability, and acceptability of collecting SDM appraisal information from patients to, eventually, the utility of real-time SDM feedback to surgeons in improving SDM with surgical patients.
--	--

REVIEWER	Swart, Michael Torbay Hospital, Anaesthesia
REVIEW RETURNED	22-Sep-2023

GENERAL COMMENTS	I think this looks like a great project. I support it and have only ticked the minor revision box for the following two thoughts and one question. First thought and a question. I was unclear at which stages on a surgical pathway the Shared Decision Making (SDM) was going to be assessed. There are multiple contacts between patients and health care professionals where SDM and even changes in decision take place as a patient progresses down a surgical pathway. The role of an MDT and the communication between clinicians also impacts on decisions. For some patients this can be complex. Are you assessing decisions or consultations? Second thought about including decisional regret at some point in your project. Is it worth looking at a simple decision regret question after surgery: "are you pleased or do you regret the decision to have surgery?" Then compare this with your SDM assessments. Good luck Mike Swart
---

VERSION 1 – AUTHOR RESPONSE

Response to reviewer comments

Reviewer: 1
Dr. Angela Bader, Brigham and Women's Hospital
Comments to the Author:

Comment 1.1

I appreciate the opportunity to review this interesting manuscript, however would need more information to accurately comment on the proposed protocol.

1.First, no degrees are listed with the authors. What exactly are the degrees, are any of the authors in clinical departments, and are any of the coauthors anesthesiologists or surgeons? I think clinical expertise [particularly in surgical areas is required among the authors, not just among those interviewed.

Response: We thank the reviewers for suggesting these additions to the author information. We have now updated the author list with professional roles and degrees for transparency about clinical expertise, demonstrating involvement of a wide range of clinical and surgical specialties (incl. anaesthesiology). For consistency, we have added information about professional roles for all authors.

Comment 1.2

2. There needs to be tables and figures of exactly what the survey questions are that are going to be sent, are three different surveys actually going to be sent to patients? will they be combined? I don't think I can comment without an understanding of exactly what is going to be sent to the patients and how the answers are going to be recorded. There is a sentence that states "SDM patient surveys will be set via SMS or email within one day after booking" as well as the day before surgery. In my own institution this would result in an extremely low response rate, have the authors considered this? Have they done any small pilots to see if this would even be feasible? I cannot tell from the methods section exactly what is going to be pulled from the EMR to answer the questions asked. Have the SDM instruments actually been used for surgical patients successfully to obtain the exact information that these authors are seeking? The references quoted dont give this specific information.

Response: We thank the reviewer for highlighting this and we agree that more information is needed here.

To provide further clarification about the study flow, we have also now included Figure 1 to illustrate the pathways and timepoints at which instruments will be delivered to patients. Measures will be operationalised into an online questionnaire and combined in a single survey administered at the same time. Screenshots of operationalised surveys will be provided within publications reporting the results of this study.

We have also included all full instrument measurements in a new supplementary file 1 (Figures S1-S3) and have referred to this in the main body (p.9). The CollaboRATE and SD-Q-9 measures have both been used to measure shared decision making for surgery. Relevant information is now included in the section "Measurement of patient experience" (p.9) which now reads: "The CollaboRATE instrument is a validated 3-item patient-reported measure assessing the extent of SDM experienced by patients [1]. Assessment of the instrument using COSMIN methods demonstrated acceptable discriminative validity, concurrent validity, intra-rater reliability and sensitivity to change [2]. It has been used in excess of 40 studies [3], including evaluations of quality improvement projects in surgery [4]. The SDM-Q-9 instrument is a validated 9-item patient-reported measure that evaluates their perceptions of involvement in the decision-making process [5]. It has been widely used in interventional studies and demonstrates good reliability, structural validity [2,5]. Systematic review evidence recommended use of SDM-Q-9 for surgery [6]." We have also made more explicit the justification for why the third instrument, the SHARED measure, was chosen: "This measure is included because it was developed, validated, and used in an NHS quality improvement programme [7,8] and evaluates domains beyond patient perception of professional communication."

The reviewer also makes an important comment about response rates. We are, at this stage, uncertain about the exact response rate. We expect a similar response rate to that reported in Studies evaluating similar measurement systems which employed a similar mode of administration have reported a wide range of response rates (eg, 18% [9], 20% [10], 30% [11] or 45% [12]). We therefore expect the response rate in our study to be within this range. It is also important to point out the large number of patients booked for surgery who will be eligible for the study. Scoping at one participating site (North Bristol NHS Trust) has identified approximate numbers of patients across seven participating surgical departments that will be included: A total of 24,865 patients were added to

procedure waiting lists in the hospital administration system between January and October 2020, of which 15,676 were ineligible (2,600 had no procedure, 11,937 had endoscopic procedures and 2,839 had emergency procedures as inpatients or on the same day as booking). This corresponds to approximately 919 patients per month who will receive invitations to participate in the survey. We therefore expect a large total number of responses.

Reviewer: 2

Dr. Justin Clapp, University of Pennsylvania

Comments to the Author:

Comment 2.1

Thanks for the opportunity to review this manuscript. The authors describe a protocol for measurement of shared decision making in a wide array of patients across 3 NHS trusts as the initial phase of a multi-phase project for increasing the frequency and quality of SDM. This initial phase also involves extensive mixed-methods assessment of the feasibility and acceptability of the authors' strategy for measuring SDM. The protocol is well written and well cited, though I do have a few comments/concerns.

We thank the reviewer for considering our protocol well-written and for providing insightful comments and suggestions. We have addressed these in our response below and in the main body and believe these changes greatly improved our manuscript.

Comment 2.2

(1) I find myself puzzled by the order of the phases. In 1.1, the authors appear to be launching their SDM measurement approach (consisting of 3 surveys disseminated via phone/email to patients identified through the HER) in order to assess recruitment and response rates. But then in 1.2 the authors describe carrying out usability testing. Wouldn't it be preferable to first carry out usability testing before even figuring out which surveys should be used and how they should be disseminated and formatted? Then in 1.3, the authors describe an extremely extensive plan to obtain feedback from patients and members of the public in part about the "acceptability and potential impact of real-time monitoring of SDM." But again, shouldn't this feedback be obtained prior to extensively launching a set of surveys that have already been decided upon by the investigators?

Response: Thank you for raising these important points. We added further detail to the manuscripts to address the reviewer's queries. We have added clarification as to the order of phases to explain why usability testing was completed at that stage. Specifically, the system has been used to facilitate electronic data capture for patient-reported outcomes prior to procurement for this study. It has therefore already undergone usability testing and refinement in other contexts and we therefore wish to undertake post-deployment usability testing for using the system to measure shared decision making for surgery. We have added detail which now reads: "The system is a customisable off-the-shelf electronic patient-reported outcome measurement software and has previously been used for electronic data capture in other countries." (p. 7) and "Post-deployment usability testing will be conducted according to International Standards Organisation (ISO) standards for human-systems interaction (9241-11:2018) to evaluate the system's use in this context" (p.10).

We also appreciate the reviewer recognising our work plan as extremely extensive. All surveys used in this study are validated questionnaires developed to measure shared decision making. The CollaboRATE, SDM-Q-9 and SHARED measures have all undergone extensive testing and their content validity has been demonstrated in previous work [1,2,17–19,4,5,7,8,13–16]. This protocol focusses on the usability and feasibility of the process of real-time monitoring of SDM and all work in Phase 1 is explorative. This includes exploring in-depth the issues with usability of the electronic system from the perspective of underserved groups with a view to optimise real-time measurement before its wider implementation in phase 2 for which methods will be reported in future publications.

Comment 2.3

(2) The sheer amount of work proposed by the investigators to evaluate their measurement approach and possibilities for an intervention must be emphasized. Just the qualitative work in 1.3 would exceed the amount of data in many PhD dissertations in sociology or anthropology. I understand the urge to thoroughly assess feasibility/acceptability/usability and to get the input of diverse stakeholders, but I have doubts that all of this work can be accomplished in a reasonably timely fashion, and I'm really not sure it's all necessary to design a method for pushing out some brief surveys.

Response: We appreciate the reviewer highlighting the amount of effort to complete this work. We feel that this investment is needed to address an urgent need to ensure large-scale, inclusive and sustainable measurement of shared decision making in clinical practice. We are confident to have assembled an experienced and dedicated multi-disciplinary team to complete the project to time and target. Data collection is currently ongoing and nearing completion. Additional funding has been awarded to complete work described in 1.3 and we will draw on expertise developed in the Born in Bradford programme which has a track record of designing and delivering citizen science projects [20–22].

Comment 2.4

(3) This is a big-picture comment, but I wonder about the worthwhileness of all this work given that it's all predicated on the SDM construct. We're a good 30 years into SDM work, and so much remains flimsy: as the authors note, we still lack a clear definition of what SDM is; there isn't much evidence that it can be reliably identified in practice, given that the numerous surveys and observation scales that have been designed to measure it have consistently been shown not to correlate; and there is, likely partially as a result of these problems, little evidence that SDM produces substantial effects on outcomes. I just wonder whether some portion of all this effort should instead be used to try to refine our models of high-quality clinical communication.

Response: We thank the reviewer for this insightful comment and we fully agree that clinical communication is potentially an important area to focus on. Whilst this particular area is outside the scope of this current work, we believe that high quality measurement of patients' experience of SDM in the first instance is needed to help build conclusive evidence and demonstrate effectiveness of any interventions to improve SDM. Evidence shows, for example, that SDM may enhance clinician and patient reasoning ('collaborative deliberation')[23], leading to more effective care[24]. Studies also showed that patients may have more realistic treatment expectations[25], or a reduced preference for intensive treatments (such as surgery) [26] which may lead to lower health service utilisation [27,28] and improved self-management [29]. The underlying mechanisms for improving SDM, however, are not yet fully understood. Our overall project aims to provide an understanding of how a decision support intervention using real-time monitoring can impact clinical and health service outcomes through improved SDM experience.

Comment 2.5

(4) Two minor notes:

a. The authors repeatedly state that they will be performing “real-time” measurement of SDM, but I was never able to figure out what exactly that means (i.e., what sets their approach off from any other effort to measure SDM in the clinic).

Response: Thank you for pointing out that further clarification is needed to define real-time measurement. We have illustrated in a newly added illustration (Figure 1) the “real-time” components of the project which are activities and processes happening within 24 hours. We have also made reference to these components in the section “Measurement of patient experience”(p. 9) which now reads: “The automated system will send three validated SDM measurement instruments within one day after surgery booking (i.e. real-time baseline measurement). The selected measurement instruments will be operationalised into an online survey and administered via short messaging service (SMS) or email. A reciprocal data feed will securely return patients’ survey responses immediately to the hospital data warehouse for secure storage (real-time analysis and feedback). Follow-up measures are sent within one day before surgery by either SMS or email (real-time follow-up measurement). A schematic of the process and intervention aims is illustrated in Figure 1.”

Comment 2.6

b. I don’t have a clear sense of how the authors will be obtaining consent from the patients whose info is apparently being derived from the EHR so that the surveys can be pushed out to them.

Response: Thank you for this important query and we have provided further clarification in the Ethics and dissemination section of the manuscript. The additional information about consent to participate in surveys reads “Patients will provide consent to participate in real-time monitoring through indicating their agreement with Terms and Conditions for the programme of work before completing the survey administered through the measurement system.”

For consistency, we have also included information about additional consent processes for collecting qualitative data collection in the same section. This now reads: “Participants will provide written consent to participate in qualitative data collection before any research activity will commence.”

Reviewer: 3

Dr. Sabrina Sanchez, Boston Medical Center

Comments to the Author:

Comment 3.1

Thank you for the opportunity to review this manuscript. First, I would like to commend the authors on a herculean undertaking- the improvement of shared decision making use in surgery is as important and timely as it is challenging. I specially appreciate that the authors are starting small and right at the begging by creating a protocol to evaluate the feasibility, usability, and acceptability of the first steps of an intervention they envision will help increase the use and improve the process of SDM in surgery. They approached their intervention design thoughtfully, methodically, comprehensively, and most importantly, humbly with regards to all the stakeholders involved in the process.

Response: We thank the reviewer for their endorsement of our work and supporting our methodological approaches. We value the reviewer's feedback to improve our manuscript and have addressed all comments as explained in our responses below.

Comment 3.2

I only have some clarifying questions with regards to the protocol that once addressed I think will strengthen the manuscript for thorough understanding by the readers:

1- Page 8, line 51: Patients without capacity to consent for medical procedures will be excluded, how will this be determined? and/or, at what point in the protocol will this be determined? Will these patients receive the SDM instruments to fill out and once it is determined they are unable to consent, be excluded? Or will they be excluded beforehand, through the algorithm developed with the software provider?

Response: Many thanks for highlighting this and we can confirm that eligibility will be determined through routinely collected patient data. We have added clarification in the "Participants" section which now states: "Data related to eligibility criteria are routinely collected through electronic patient record (EPR) systems."

In the "Measurement of patient experience" section, we have stated that "Eligible patients will be identified through routine electronic patient records using algorithms developed in collaboration with the software provider. Structured data queries will be designed to extract details of patients booked for eligible procedures." To explain that eligibility will be determined automatically through EPR data queries before surveys will be sent.

Comment 3.3

2- Page 9, line 15: I am curious about the booking process after a patient sees a surgeon. In order to have this intervention really be a "real time" intervention, it would be important to ensure the SDM questionnaires are received/filled out by the patient shortly after the decision-making conversation with the surgeon (hours to days). It seems like the algorithm will be such that a patient will receive the SDM questionnaires after the surgery is booked in the EPR, so, at the participating hospitals, are surgeries always booked within hours to days of the decision for surgery being made? At many of the hospitals that I know of, that is not the case, and very often cases don't get booked for weeks to months after the decision for surgery is made. In hospitals of this kind the intervention presented would not be "real time". Can you please clarify this point?

Response: Thank you for these helpful comments. We have provided further detail about the assumptions underlying the chosen measurement timepoint in response to other reviewers' queries (see response to comments 1.2 and 4.1) and have included Figure 1 to further illustrate the process by which real-time measurement of SDM is intended (see p. 9).

In the context of the UK National Health service, surgery bookings are only performed upon an explicit decision made by the patient, usually during consultations. Demand for surgery is managed through surgery waiting lists (often managed by a waiting list co-ordinator) and waiting times can vary significantly (from weeks to years, depending on condition and urgency). It can be assumed that in most cases, surgery booking is being made by the waiting list coordinator promptly after decisions for surgery have been communicated. Nevertheless, the reviewer makes an important point in that the time between surgery booking and the date of surgery can vary considerably depending on specialty, condition and chosen treatment. This is, to our knowledge, the first study to measure shared decision making at large scale and we are at this stage uncertain about the impact of surgery waiting times and hope to explore this in this study, in particular in work packages 1.1 and 1.3 when exploring

feasibility and impact of SDM measurement. We therefore also included a large range of elective and urgent services in our study to explore this issue in detail.

Comment 3.4

3- Page 9, lines 49-51: Will you evaluate the time frame of responses after patients receive the questionnaire? It seems like this may be an important thing to measure with regards to evaluating the feasibility of this project. It may be very different if someone responds within 3 hours of receiving the surveys vs 3 weeks. If you are able to get a 90% response rate but you only get this several weeks after the initial conversation, this may not be as useful as getting a 90% response rate at 3 days, and a 3 week delay would not constitute "real time", while 3 days could

Response: Thank you for this helpful suggestion and we agree that investigating time to response will be of value. We have therefore included this metric in the "Analysis" section of 1.1 Feasibility of automated data collection (p.10) to be examined alongside response rates.

Comment 3.5

4- On a similar note, do you have an idea yet of what "real time" may mean in the context of this intervention? Will it be a few days, a week, a month? Or is this something that you will gather based on responses once the intervention starts?

Response: Thank you. Another reviewer posed the same question and we would therefore like to refer the reviewer to our response to comment 2.6 made by reviewer 2, we have also provided further explanation about the consent processes and have added relevant detail to the "Ethics and dissemination" section.

Comment 3.6

5- Page 12, line 54: A reference or two supporting that economically disadvantaged, minority race/ethnic groups, and elderly patients are disproportionately affected by poor SDM processes would be helpful.

Response: Again, thank you for this helpful comment. We have now included the following four references to support this statement:

- 48 Chen JC, Obeng-Gyasi S. Intersectionality and the Surgical Patient: Expanding the Surgical Disparities Framework. *Ann Surg* 2022;**275**:E3–5. doi:10.1097/SLA.0000000000005045
- 49 Durand MA, Carpenter L, Dolan H, *et al.* Do interventions designed to support shared decision-making reduce health inequalities? A systematic review and meta-analysis. *PLoS One* 2014;**9**. doi:10.1371/JOURNAL.PONE.0094670
- 50 Tarver WL, Haggstrom DA. The Use of Cancer-Specific Patient-Centered Technologies Among Underserved Populations in the United States: Systematic Review. *J Med Internet Res* 2019;**21**:e10256. doi:10.2196/10256
- 51 De Acedo Lizárraga MLS, De Acedo Baquedano MTS, Cardelle-Elawar M. Factors that affect decision making: Gender and age differences. *Int J Psychol Psychol Ther* 2007;**7**:381–91. <https://www.redalyc.org/pdf/560/56070306.pdf> (accessed 1 Nov 2023).

Comment 3.7

6- I eventually understood that you will be rolling out the SDM questionnaires as part of QI and as such, all patients that meet inclusion criteria will be included, there will be no patient consent required, and there will be no option to not participate as you evaluate the feasibility of automated data collection. While this is explained at the end of the manuscript I spent several pages prior, as I was reading, wondering about consent and opt out ability for patients from even receiving the questionnaires. I feel like it may be helpful to explain this explicitly and early in the manuscript (potentially the "Setting" section).

Response: We thank the reviewer for this suggestion and have followed their recommendation to add detail in the "setting" section. The first sentence now reads: Research will be conducted at three UK hospital trusts [...] alongside quality improvement programmes to improve SDM." In our response to comment

Comment 3.8

Again, this is an incredibly timely and important step in improving SDM in surgery and I am eager to see the results of this intervention at all stages, from the feasibility, usability, and acceptability of collecting SDM appraisal information from patients to, eventually, the utility of real-time SDM feedback to surgeons in improving SDM with surgical patients.

Response: Once again, many thanks for the positive feedback, helpful suggestions to improve the methods and supporting our project plans.

Reviewer: 4

Dr. Michael Swart, Torbay Hospital

Comments to the Author:

Comment 4.1

I think this looks like a great project. I support it and have only ticked the minor revision box for the following two thoughts and one question. First thought and a question. I was unclear at which stages on a surgical pathway the Shared Decision Making (SDM) was going to be assessed. There are multiple contacts between patients and health care professionals where SDM and even changes in decision take place as a patient progresses down a surgical pathway. The role of an MDT and the communication between clinicians also impacts on decisions. For some patients this can be complex. Are you assessing decisions or consultations?

Response: The reviewer makes an important observation and we agree that further information is needed. We can confirm that this project will measure the process of SDM, rather than a single consultation which is reflected in the overall project aim: "We will develop, pilot, and evaluate a decision support intervention that uses real-time feedback of patient experiences of the SDM process to impact patient and professional decision-making processes before adult elective surgery and improve patient and health service outcomes"

We have made amendments throughout the manuscript to write "experience of the SDM process" and have added further clarification in the "Measurement of patient experience" section on p.9 in line with the reviewer's observation: "This timepoint in the decision making process was chosen as a pragmatic point in time to represent patients' cumulative experiences of SDM for surgery which may include discussions with surgeons, physicians, general practitioners, nurses, family, and friends."

Comment 4.2

Second thought about including decisional regret at some point in your project. Is it worth looking at a simple decision regret question after surgery: "are you pleased or do you regret the decision to have surgery?" Then compare this with your SDM assessments. Good luck Mike Swart

Response: Thank you for this suggestion and we agree that decision regret is an interesting outcome to investigate. A large body of research, however, has already demonstrated the relationship between high-quality SDM and decision regret for surgery [30]. We agree there is value in demonstrating evidence for improvements in patient and health service outcomes. The selection of a primary outcome measure for intervention development and evaluation, however, will be determined with multi-stakeholder input (Phase 2) and reported in future publications. In response to comment 2.4, we also discussed the potential for this work to aid our understanding of underlying mechanisms for various benefits of high-quality SDM.

References

- 1 Elwyn G, Barr PJ, Grande SW, *et al.* Developing CollaboRATE: A fast and frugal patient-reported measure of shared decision making in clinical encounters. *Patient Educ Couns* 2013;**93**:102–7. doi:10.1016/J.PEC.2013.05.009
- 2 Gärtner FR, Bomhof-Roordink H, Smith IP, *et al.* The quality of instruments to assess the process of shared decision making: A systematic review. *PLoS One* 2018;**13**. doi:10.1371/journal.pone.0191747
- 3 Scholl I, Loon MK Van, Sepucha K, *et al.* Measurement of shared decision making – a review of instruments. *Z Evid Fortbild Qual Gesundheitswes* 2011;**105**:313–24. doi:10.1016/J.ZEFQ.2011.04.012
- 4 Brodney S, Fowler FJ, Barry MJ, *et al.* Comparison of Three Measures of Shared Decision-Making: SDM Process_4, CollaboRATE, and SURE Scales. *Med Decis Making* 2019;**39**:673. doi:10.1177/0272989X19855951
- 5 Kriston L, Scholl I, Hölzel L, *et al.* The 9-item Shared Decision Making Questionnaire (SDM-Q-9). Development and psychometric properties in a primary care sample. *Patient Educ Couns* 2010;**80**:94–9. doi:10.1016/J.PEC.2009.09.034
- 6 de Mik SML, Stubenrouch FE, Balm R, *et al.* Systematic review of shared decision-making in surgery. *Br J Surg* 2018;**105**:1721–30. doi:10.1002/bjs.11009
- 7 NHS Rightcare. Measuring Shared Decision Making: A review of research evidence - A report for the Shared Decision Making programme in partnership with Capita Group Plc. London: 2012. <https://www.england.nhs.uk/wp-content/uploads/2013/08/7sdm-report.pdf>
- 8 NHS Rightcare. Your Health – Your Decision. London: 2013. <https://aqua.nhs.uk/wp-content/uploads/2021/10/Your-Health-Your-Decision-Evaluation-Report.pdf>
- 9 Iversen HH, Holmboe O, Bjertnaes O. Patient-reported experiences with general practitioners: a randomised study of mail and web-based approaches following a national survey. *BMJ Open* 2020;**10**:e036533. doi:10.1136/BMJOPEN-2019-036533
- 10 Bliddal S, Banasik K, Pedersen OB, *et al.* Acute and persistent symptoms in non-hospitalized PCR-confirmed COVID-19 patients. *Sci Reports* 2021 *111* 2021;**11**:1–11. doi:10.1038/s41598-021-92045-x
- 11 Arner M. Developing a national quality registry for hand surgery: challenges and opportunities. *EFORT Open Rev* 2016;**1**:100–6. doi:10.1302/2058-5241.1.000045
- 12 Warwick H, Hutyra C, Politzer C, *et al.* Small Social Incentives Did Not Improve the Survey Response Rate of Patients Who Underwent Orthopaedic Surgery: A Randomized Trial. *Clin Orthop Relat Res* 2019;**477**:1648. doi:10.1097/CORR.0000000000000732
- 13 Barr PJ, Thompson R, Walsh T, *et al.* The Psychometric Properties of CollaboRATE: A Fast and Frugal Patient-Reported Measure of the Shared Decision-Making Process. *J Med Internet Res* 2014;**16**(1)e2 <https://www.jmir.org/2014/1/e2> 2014;**16**:e3085. doi:10.2196/JMIR.3085
- 14 Ubbink DT, van Asbeck E V., Aarts JWM, *et al.* Comparison of the CollaboRATE and SDM-Q-9 questionnaires to appreciate the patient-reported level of shared decision-making. *Patient Educ Couns* 2022;**105**:2475–9. doi:10.1016/J.PEC.2022.03.007
- 15 Doherr H, Christalle E, Kriston L, *et al.* Use of the 9-item Shared Decision Making

- Questionnaire (SDM-Q-9 and SDM-Q-Doc) in intervention studies—A systematic review. *PLoS One* 2017;**12**. doi:10.1371/JOURNAL.PONE.0173904
- 16 De las Cuevas C, Mundal I, Betancort M, *et al*. Assessment of shared decision-making in community mental health care: Validation of the CollaboRATE. *Int J Clin Heal Psychol* 2020;**20**:262–70. doi:10.1016/J.IJCHP.2020.06.004
 - 17 Hurley EA, Bradley-Ewing A, Bickford C, *et al*. Measuring shared decision-making in the pediatric outpatient setting: Psychometric performance of the SDM-Q-9 and CollaboRATE among English and Spanish speaking parents in the US Midwest. *Patient Educ Couns* 2019;**102**:742–8. doi:10.1016/J.PEC.2018.10.015
 - 18 Durand MA, Bekker HL, Casula A, *et al*. Can we routinely measure patient involvement in treatment decision-making in chronic kidney care? A service evaluation in 27 renal units in the UK. *Clin Kidney J* 2016;**9**:252–9. doi:10.1093/CKJ/SFW003
 - 19 Bekker H, Winterbottom A, Gavaruzzi T, *et al*. Decision Aids to Assist Patients, and Professionals, in Choosing the Right Treatment for Kidney Failure (in press). *Clin Kidney J* 2023.
 - 20 Rahman A, Nawaz S, Khan E, *et al*. Nothing about us, without us: is for us. *Res Involv Engagem* 2022;**8**:1–10. doi:10.1186/S40900-022-00372-8/FIGURES/1
 - 21 Dickerson J, Bird PK, Bryant M, *et al*. Integrating research and system-wide practice in public health: Lessons learnt from Better Start Bradford. *BMC Public Health* 2019;**19**:1–12. doi:10.1186/S12889-019-6554-2/TABLES/8
 - 22 Islam S, Joseph O, Chaudry A, *et al*. “We are not hard to reach, but we may find it hard to trust” Involving and engaging ‘seldom listened to’ community voices in clinical translational health research: a social innovation approach. *Res Involv Engagem* 2021 71 2021;**7**:1–15. doi:10.1186/S40900-021-00292-Z
 - 23 Elwyn G, Lloyd A, May C, *et al*. Collaborative deliberation: A model for patient care. *Patient Educ Couns* 2014;**97**:158–64. doi:10.1016/J.PEC.2014.07.027
 - 24 Elwyn G, Frosch DL, Kobrin S. Implementing shared decision-making: Consider all the consequences. *Implement Sci* 2016;**11**:1–10. doi:10.1186/S13012-016-0480-9/TABLES/2
 - 25 Lantz PM, Janz NK, Fagerlin A, *et al*. Satisfaction with Surgery Outcomes and the Decision Process in a Population-Based Sample of Women with Breast Cancer. *Health Serv Res* 2005;**40**:745–68. doi:10.1111/J.1475-6773.2005.00383.X
 - 26 Langseth MS, Shepherd E, Thomson R, *et al*. Quality of decision making is related to decision outcome for patients with cardiac arrhythmia. *Patient Educ Couns* 2012;**87**:49–53. doi:10.1016/J.PEC.2011.07.028
 - 27 Hughes TM, Merath K, Chen Q, *et al*. Association of shared decision-making on patient-reported health outcomes and healthcare utilization. *Am J Surg* 2018;**216**:7–12. doi:10.1016/J.AMJSURG.2018.01.011
 - 28 Arterburn D, Wellman R, Westbrook E, *et al*. Introducing decision aids at group health was linked to sharply lower hip and knee surgery rates and costs. *Health Aff* 2012;**31**:2094–104. doi:10.1377/hlthaff.2011.0686
 - 29 Heisler M, Bouknight RR, Hayward RA, *et al*. The Relative Importance of Physician Communication, Participatory Decision Making, and Patient Understanding in Diabetes Self-

management. *J Gen Intern Med* 2002;**17**:243–52. doi:10.1046/J.1525-1497.2002.10905.X

- 30 Wilson A, Ronnekleiv-Kelly SM, Pawlik TM. Regret in Surgical Decision Making: A Systematic Review of Patient and Physician Perspectives. *World J Surg* 2017;**41**:1454–65. doi:10.1007/S00268-017-3895-9/FIGURES/2

VERSION 2 – REVIEW

REVIEWER	Clapp, Justin University of Pennsylvania
REVIEW RETURNED	30-Nov-2023

GENERAL COMMENTS	I commend the authors for their thorough responses. I'm satisfied with the changes.
---

REVIEWER	Sanchez, Sabrina Boston Medical Center
REVIEW RETURNED	20-Nov-2023

GENERAL COMMENTS	I appreciate the author's revisions of the manuscript and again commend them on their efforts to improve SDM for surgical patients in the UK. The following are editing suggestions to address grammatical errors and inconsistencies in writing; they in no way affect the content of the manuscript, which I do not believe needs any further revisions. Page 5, Line 20: I would specify that you are referring to "decisional capacity" Page 6, Line 47-49: Consider numbering the priority areas ("...how to: 1) sustain..., 2) measure..., and 3) ensure the SDM process...") to make it easier to read/follow Page 10, Line 19: For consistency, remove the "i.e." Page 11, Line 12-13: "... overall recruitment rate, response rates and time to response...." Page 11, Line 26: Remove "will be explored" Box 1: The 3rd point has an extra "is" ("User satisfaction: the subjective...") Page 13, Line 59: "... with a wide range of stakeholders..." Page 14, Line 8: Capitalizing the P in "phase 2" Page 15, Line 7/8: "...will be required..." Page 17, Line 27: "... consisting of members..." Page 17, Line 31: "... will be coordinated by..." Overall:  - Ensure you use either i.e. or e.g. throughout - After point 1.1 and 1.3 you have subheadings marked just by italicizing the text, while in point 1.2 you denote subheadings as 1.2.1, 1.2.2, etc. Modify for consistency - When listing points in the text use 1, 2, 3, etc, or use i, ii, iii, etc, but not both (see page 15, line 3, for example)
---

VERSION 2 – AUTHOR RESPONSE

Reviewer: 3

Dr. Sabrina Sanchez, Boston Medical Center

Comments to the Author:

I appreciate the author's revisions of the manuscript and again commend them on their efforts to improve SDM for surgical patients in the UK.

The following are editing suggestions to address grammatical errors and inconsistencies in writing; they in no way affect the content of the manuscript, which I do not believe needs any further revisions.

Page 5, Line 20: I would specify that you are referring to "decisional capacity"

Page 6, Line 47-49: Consider numbering the priority areas ("...how to: 1) sustain..., 2) measure..., and 3) ensure the SDM process...") to make it easier to read/follow

Page 10, Line 19: For consistency, remove the "i.e."

Page 11, Line 12-13: "... overall recruitment rate, response rates and time to response...."

Page 11, Line 26: Remove "will be explored"

Box 1: The 3rd point has an extra "is" ("User satisfaction: the subjective...")

Page 13, Line 59: "... with a wide range of stakeholders..."

Page 14, Line 8: Capitalizing the P in "phase 2"

Page 15, Line 7/8: "...will be required..."

Page 17, Line 27: "... consisting of members..."

Page 17, Line 31: "... will be coordinated by..."

Overall:

- Ensure you use either i.e. or e.g. throughout
- After point 1.1 and 1.3 you have subheadings marked just by italicizing the text, while in point 1.2 you denote subheadings as 1.2.1, 1.2.2, etc. Modify for consistency
- When listing points in the text use 1, 2, 3, etc, or use i, ii, iii, etc, but not both (see page 15, line 3, for example)

Response: We thank the reviewer for the supportive comments about our project and we appreciate the helpful review and suggestions to improve the writing and formatting of our manuscript. We have addressed all the editing suggestions made by the reviewer.